- Quantitative reconstruction of deglacial bottom-water nitrate in marginal Pacific
- seas using the pore density of denitrifying benthic foraminifera
- Anjaly Govindankutty Menon<sup>1\*</sup>, Aaron L. Bieler<sup>2, 3</sup>, Hanna Firrincieli<sup>1</sup>, Rachel Alcorn<sup>4</sup>, Niko
- Lahajnar<sup>1, 6</sup> Catherine V. Davis<sup>4</sup>, Ralf Schiebel<sup>2</sup>, Dirk Nürnberg<sup>5</sup>, Gerhard Schmiedl<sup>1, 6</sup>, Nicolaas
- Glock<sup>1</sup>

<sup>1</sup>Department of Earth System Sciences, Institute for Geology, Universität Hamburg, Bundesstrasse
 55, D - 20146 Hamburg, Germany

- <sup>2</sup>Climate Geochemistry Department Max Planck Institute for Chemistry, Hahn-Meitner-Weg 1,
- *D 55128 Mainz, Germany*

- <sup>3</sup>Department of Earth and Planetary Sciences, ETH Zürich, Sonneggstrasse 5,
- 14 8092 Zürich, Switzerland

- <sup>4</sup>Department of Marine, Earth, and Atmospheric Sciences, North Carolina State University, 2800
- 17 Faucette Dr, Raleigh, NC, 27607, United States

- <sup>5</sup>GEOMAR Helmholtz Centre for Ocean Research Kiel, Wischhofstr. 1-3, Geb. 8c, Raum 106, D-
- *24148 Kiel, Germany*

- <sup>6</sup>Center for Earth System Research and Sustainability, Institute for Geology, Universität Hamburg,
- 23 Bundesstrasse 55, D-20146 Hamburg, Germany

\*Corresponding author: anjalygmenon9@gmail.com

#### **Abstract**

Quantifying past ocean nitrate concentrations is crucial for understanding the global nitrogen cycle. Here, we reconstruct deglacial bottom-water nitrate concentrations ([NO<sub>3</sub>-]<sub>BW</sub>) in the oxygen-deficient zones of the Sea of Okhotsk, the Gulf of California, the Mexican Margin, and the Gulf of Guayaquil. Using the pore density of denitrifying benthic foraminifera as a nitrate proxy, differences in [NO<sub>3</sub>-]<sub>BW</sub> are observed at the study sites spanning the Last Glacial Maximum to the Holocene. Changes in water-column denitrification, water-mass ventilation, primary productivity, and sea surface temperatures may account for nitrate differences at the study sites. The [NO<sub>3</sub>-]<sub>BW</sub> in the Sea of Okhotsk, the Gulf of California, and the Gulf of Guayaquil are influenced by the intermediate water masses while, the [NO<sub>3</sub>-]<sub>BW</sub> at the Mexican Margin is likely influenced by deglacial changes in the Pacific Deep Water. The comparison of past and present [NO<sub>3</sub>-] shows that the modern Gulf of Guayaquil and the Gulf of California currently have stronger oxygen-deficient zones with higher denitrification than during the Last Glacial Maximum. In contrast, the modern Mexican Margin and the Sea of Okhotsk may have higher oxygen than during the Last Glacial Maximum, indicated by low modern denitrification.

#### 1. Introduction

75

The marine nitrogen cycle is a complex web of microbially mediated processes controlling the 76 inventory and distribution of bioavailable nitrogen in marine environments (Casciotti, 2016). 77 Biological nitrogen fixation by nitrogen-fixing diazotrophs (e.g., cyanobacteria) in the surface 78 79 layer is the main source of bioavailable nitrogen in the ocean, and denitrification and anammox, are the main fixed nitrogen loss processes (Lam and Kuypers, 2011), both of which occur under 80 low-oxygen conditions. The primary form of bioavailable nitrogen in the ocean is nitrate (NO<sub>3</sub><sup>-</sup>), 81 (Casciotti 2016), which is a limiting nutrient throughout the tropical and subtropical oceans (Moore 82 et al., 2013). 83 Oxygen-deficient zones (ODZs) are regions of very low dissolved oxygen (O2) where the O2 84 85 concentration is less than 22 µmol/kg, usually within depths of 100-1,200 m (Levin, 2003; 2018). Oxygen plays a key role in the marine nitrogen cycle (Keeling et al., 2010) because some microbial 86 87 processes require oxygen while others are inhibited by it (Voss et al., 2013). For example, denitrification (reduction of nitrate to dinitrogen gas) in the ocean occurs in suboxic (oxygen 88 89 <5μmol/kg) conditions (Codispoti et al., 2001; Levin, 2018). On a global scale, ~30-50% of fixed nitrogen loss in the world's oceans occurs in ODZs (Gruber, 2008), either through denitrification 90 91 or anammox (Devol et al., 2006; Lam and Kuypers, 2011; Evans et al., 2023). Due to the complex interactions and feedbacks within the biogeochemical nitrogen cycle, the amount of benthic 92 denitrification also influences other important processes, such as global nitrogen fixation and net 93 primary production (Somes et al., 2017; Li et al., 2024). Oxygen Deficient Zones cover only 1% 94 of the world's seafloor (Codispoti et al., 2001), however, 10% of the global benthic denitrification 95 96 occurs in these regions (Bohlen et al., 2012). Observations and climate model simulations have predicted that ODZs will continue to expand until at least the year 2100 (Stramma et al., 2008, 97 98 2010; Schmidtko et al., 2017; Oschlies, 2021). However, the long-term evolution of ODZs remains uncertain (Yamamoto et al., 2015; Takano et al., 2018; Fu et al., 2018; Frölicher et al., 2020). 99 100 There is growing evidence that ODZs may contract during transient and equilibrium climate warmings over timescales of millennia and beyond (Auderset et al., 2022; Moretti et al., 2024). 101 Considering the role of ODZs in modulating the marine nitrogen cycle, it is of key scientific 102 interest to understand how nitrogen cycling works in these ecosystems and the potential factors 103 104 that influence the nitrogen cycle.

In this study, we use the pore density (number of pores per unit area) of Bolivina spissa and 105 Bolivina subadvena as a NO<sub>3</sub> proxy (Fig. 1 (a)) to reconstruct bottom-water nitrate [NO<sub>3</sub>]<sub>BW</sub> in 106 107 intermediate water depths of the Sea of Okhotsk, the Gulf of California, the Gulf of Guayaquil, and in the Pacific Deep Water (PDW) depths of the Mexican Margin (Fig. 2 and 3). The [NO<sub>3</sub>]<sub>BW</sub> 108 calibration using the pore density of B. spissa and B. subadvena (see Fig. 1 (b)) developed in 109 Govindankutty Menon et al. (2023) is applied in the current study. Combining a proxy for 110 [NO<sub>3</sub>]<sub>BW</sub> (pore density of denitrifying foraminifera) and a proxy for N-cycle processes in the 111 water column (δ<sup>15</sup>N<sub>bulk</sub>) facilitates a more comprehensive understanding of past N-cycling in 112 different zones of the water column. Here, we try to understand 1) whether there are differences 113 in reconstructed [NO<sub>3</sub>]<sub>BW</sub> between today, deglacial, and glacial periods in the four studied sites, 114 and 2) whether the reconstructed [NO<sub>3</sub><sup>-</sup>]<sub>BW</sub> records are in agreement with insights drawn from 115  $\delta^{15}N_{bulk}$  data. 116

# 1.1 Application of $\delta^{15}N_{bulk}$ and its potential limitations

117

The stable isotope signature of nitrogen in the sedimentary organic matter ( $\delta^{15}N_{bulk}$ ) is an 118 established proxy for water-column denitrification and for understanding changes associated with 119 120 nutrient utilization (Thunell et al., 2004; Robinson et al., 2009; Martinez and Robinson, 2010; Dubois et al., 2011, 2014; Tesdal et al., 2013; Wang et al., 2019; Riechelson et al., 2024). An 121 increase (or decrease) in nutrient availability in relation to nutrient demand results in an increase 122 (or decrease) in  $\delta^{15}$ N values (Wada and Hattori, 1978; Montova, 1990). When the oxygen in the 123 124 ocean is depleted, either due to global warming or increased remineralization, denitrification rates in the water column are also increasing and so is  $\delta^{15}N$  (Wang et al., 2019). Therefore,  $\delta^{15}N_{bulk}$  can 125 126 be an important tool for reconstructing past changes in denitrification in the ODZs.

The  $\delta^{15}N$  records from the bulk sediment can be subject to interlinked processes/or sources which can complicate their interpretation. For example, diagenetic alteration during sinking in the water column and burial in the sediment (Altabet and Francois, 1994; Lourey et al., 2003), as well as terrestrial or shelf sources of organic and inorganic nitrogen (Schubert & Calvert, 2001; Kienast et al., 2005; Meckler et al., 2011), and remotely advected water masses with different  $\delta^{15}N$  values (for e.g., Southern Californian margin; Liu and Kaplan, 1989), could influence the  $\delta^{15}N$  signatures in sediments. Nevertheless, Tesdal et al. (2013) proposed that  $\delta^{15}N_{\text{bulk}}$  can be a reliable indicator for individual locations reflecting the oceanographic conditions of the surrounding environments.

The nitrogen isotopes of organic matter bound and protected within the calcite shell of planktic 135 foraminifera ( $\delta^{15}N_{FB}$ ) are less subjected to diagenesis or sedimentary contamination than  $\delta^{15}N_{bulk}$ 136 137 and can be used to understand major nitrogen transformations occurring in the ocean (Ren et al., 2012; Studer et al., 2021). There are well-documented disagreements between bulk sediment δ<sup>15</sup>N 138 and foraminifera-bound  $\delta^{15}N$  records, particularly in glacial-interglacial comparisons (Studer et 139 140 al., 2021). While  $\delta^{15}$ N<sub>bulk</sub> suggests strong variability in water-column denitrification between the LGM and deglaciation,  $\delta^{15}N_{FB}$  records indicate a more moderate change, with a peak during 141 deglaciation but relatively stable values during the LGM and Holocene. This highlights that 142  $\delta^{15}N_{bulk}$  and  $\delta^{15}N_{FB}$  may reflect different aspects of the nitrogen cycle (Studer et al., 2021). Recent 143 studies (Auderset et al., 2022; Hess et al., 2023; Moretti et al., 2024) based on  $\delta^{15}N_{FB}$  have shown 144 that water column denitrification decreased and ODZs contracted during warmer-than-present 145 periods of the Cenozoic. In contrast, Riechelson et al. (2024) used  $\delta^{15}N_{bulk}$  and hypothesized that 146 the decrease in  $\delta^{15}N_{bulk}$  values over the Holocene is related to a decrease in Southern Ocean nutrient 147 utilization and not due to a decrease in denitrification. 148

# 1.2 Pore density of benthic foraminifera as a bottom-water nitrate proxy

Foraminifera account for a major part of benthic denitrification in the ODZs (up to 100% in some environments) (Piña-Ochoa et al., 2010a; 2010b; Glock et al., 2013; Dale et al., 2016, Chocquel et al., 2021, Rakshit et al., 2025). Some species, for example B. spissa, which are abundant in ODZs in and around the Pacific Ocean (Glock et al., 2011; Fontanier et al., 2014) can use NO<sub>3</sub> as an electron acceptor (see Fig. 1 (a)) and thus can denitrify (Risgaard-Petersen 2006; Piña-Ochoa et al., 2010a; 2010b). A study by Glock et al. (2019) proposed for some denitrifying foraminifera, denitrification is their preferred respiration pathway. The uptake of NO<sub>3</sub><sup>-</sup> by these foraminifera is likely through pores in the test (see Fig. 1 (a)). Nitrate is completely denitrified to dinitrogen gas (N<sub>2</sub>) partly by the foraminifera themselves (Risgaard-Petersen 2006; Woehle and Roy et al., 2018; Orsi et al., 2020; Gomaa et al., 2021), and partly supported by prokaryotic endobionts (Bernhard et al., 2012a, Woehle and Roy et al., 2022). To date, benthic foraminifera are the only eukaryote holobiont known to perform complete heterotrophic denitrification (Risgaard-Petersen 2006; Kamp et al., 2015). Every Bolivina species tested so far (including Bolivina seminuda), can denitrify (Piña-Ochoa et al., 2010a; Bernhard et al., 2012b), suggesting that denitrification is a common survival strategy of Bolivinidae under oxygen-depleted conditions (Glock et al., 2019). This makes species of this genus particularly suitable candidates for reconstructing past nitrate levels using pore characteristics as a proxy. In low-oxygen environments, such as the ODZs off Peru, Costa Rica, and the hypoxic Sagami Bay, *B. spissa* increase their pore density with decreasing ambient NO<sub>3</sub><sup>-</sup> availability (Govindankutty Menon et al., 2023). Thus, the pore density of several *Bolivina* species, such as *B. spissa*, and *B. subadvena*, is an empirically calibrated proxy that shows the strongest correlation with the bottom-water nitrate concentration (see Fig. 1 (b)) (Glock et al., 2011; Govindankutty Menon et al., 2023) rather than bottom-water oxygen, temperature, water depth, salinity or pore water nitrate.

Figure 1: The (a) schematic view of nitrate (NO<sub>3</sub><sup>-</sup>) uptake, and the excretion of nitrogen gas (N<sub>2</sub>) by the benthic foraminifera *Bolivina spissa*. The step-wise denitrification pathway from NO<sub>3</sub><sup>-</sup> to N<sub>2</sub> involving enzymes such as nitrate reductase (Nr Nar), nitrite reductase (Nir), nitric-oxide reductase (Nor), and nitrous oxide reductase (Nos) is also shown. (b) Correlation between pore

Figure 2. Location of sediment cores used in the current study and mean annual oxygen concentrations at 700 m depth (Garcia et al., 2019). Sediment cores are indicated by yellow triangles: Sea of Okhotsk (core MD01-2415; water depth: 822 m), Gulf of California (DSDP Site-480; water depth: 747 m), Mexican Margin (core MAZ-1E-04; water depth: 1463 m), and Gulf of Guayaquil (core M77/2-59-01; water depth: 997 m). Map created with Ocean Data View (Schlitzer, R., 2023).

Figure 3. Modern a) salinity and b) nitrate distribution along a N-S transect across the Pacific (Garcia et al., 2019) with major subsurface and deep-water masses (blue arrows) and formation areas of North Pacific Intermediate Water (NPIW) and Southern Ocean Intermediate Water (SOIW) are included. Sediment cores used for [NO<sub>3</sub>]<sub>BW</sub> reconstruction are shown (red crosses) projected to the N-S hydrographic transect. Equatorial Pacific Intermediate Water (EqPIW), Equatorial Undercurrent (EUC), NPIW, SOIW, Pacific Deep Water (PDW), Antarctic Bottom Water (AABW), and Circumpolar Deep Water (CDW). Profiles generated by Ocean Data View (Schlitzer, R., 2023) using the data from World Ocean Atlas 2018 (Garcia et al., 2019).

## 2. Materials and methods

- 2.1 Study area and sampling of sediment cores. We used downcore samples from the Eastern
- Tropical South Pacific, ETSP (Gulf of Guayaquil (M77/2-59-01), Eastern Tropical North Pacific
- the ETNP (Mexican Margin, MAZ-1E-04), the Gulf of California (Guaymas Basin, DSDP-64-
- 480), and the Sea of Okhotsk (MD01-2415), over the last ~20,000 years (Fig. 3). The Gulf of
- Guayaquil sediment core M77/2-59-01 (03°57.01′ S, 81°19.23′ W, recovery 13.59 m) was
- collected from the northern edge of the ODZ at a water depth of 997 m during the RV Meteor
- cruise M77/2 in 2008 (Mollier-Vogel et al., 2013, 2019; Nürnberg et al., 2015). The piston core
- MAZ-1E-04, Mexican Margin (22.9°N, 106.91°W) was collected on board the RV El Puma at a
- water depth of 1463 m. The CALYPSO giant piston core MD01-2415 (53°57.09' N, 149°57.52'
- E, recovery 46.23 m) was recovered from the northern slope of the Sea of Okhotsk at 822 m water
- depth during the WEPAMA cruise MD122 of the RV Marion Dufresne (Holbourn et al., 2002;
- Nürnberg & Tiedemann, 2004). The Deep-Sea Drilling Project core DSDP- 480 (27°54' N, 111°39'
- 216 W) from the Gulf of California was retrieved at a water depth of 747 m close to the Guaymas
- Basin. For details on the sampling procedure of foraminiferal specimens, please refer to the
- Supplementary Methods section in the Supplementary Information.
- 2.2 Automated image analysis. All specimens of B. spissa and B. subadvena were imaged using
- a Scanning Electron Microscope (Hitachi Tabletop SEM TM4000 series) at Hamburg University,
- Germany with an accelerating voltage of 15 kV using a back-scattered electron (BSE) detector
- (Further methodological details are provided in the Supplementary Information).
- Following the image analysis, pore density data of benthic foraminifera from the four ODZs were
- used for the quantitative reconstruction of [NO<sub>3</sub><sup>-</sup>]<sub>BW</sub> (Fig. 4). We distinguished five different time
- intervals, including the Last Glacial Maximum (LGM; 22–17 ka BP), Heinrich Stadial 1 (H1; 17–
- 15 ka BP), Bølling Allerød (BA; 14.7-12. 9 ka BP), Younger Dryas (YD; 12.9–11.7 ka BP), Early
- Holocene (EH; 11.7–8.2 ka BP) and Middle to Late Holocene (MLH; 8–0 ka BP) to describe the
- [NO<sub>3</sub><sup>-</sup>]<sub>BW</sub> in the East Pacific and the Sea of Okhotsk. We present updated chronostratigraphies of
- the studied cores, primarily based on accelerator mass spectrometry (AMS) radiocarbon (14C)
- dating, as detailed in the Supplementary Methods.
- The [NO<sub>3</sub><sup>-</sup>]<sub>BW</sub> from all cores were calculated using the calibration equation;

$$[NO_3^-]_{BW} = -3896 (\pm 350) PD + 61(\pm 1)$$
 (1)

- where PD is the pore density of benthic foraminifera (Govindankutty Menon et al., 2023).
- The standard error of the mean (SEM) for one sample was calculated using the equation;

SEM<sub>[NO<sub>3</sub>]<sub>BW</sub></sub> = 
$$\frac{SD_{[NO_3]_{BW}}}{\sqrt{n}}$$
 (2)

where n is the number of specimens analyzed in each sample and SD is 1 standard deviation of mean reconstructed [NO<sub>3</sub>-]<sub>BW</sub>.

$$SD_{[NO_3^-]_{BW}} = \sqrt{(350 X PD)^2 + (-3896 X SD_{PD})^2 + (1)^2}$$
 (3)

- A complete error propagation was done for the calculation of the errors of the reconstructed [NO<sub>3</sub>]<sub>BW</sub> including both the uncertainty of the mean PD within the samples and the uncertainties of the calibration function. The reconstructed [NO<sub>3</sub><sup>-</sup>]<sub>BW</sub> and the calculated SEM and SD of each
- sample are shown in the Supplementary Information.
- 2.3 Sedimentary nitrogen isotope ( $\delta^{15}N_{\text{bulk}}$ ) measurements. We have measured sedimentary
- nitrogen isotope ( $\delta^{15}N_{bulk}$ ) rather than  $\delta^{15}N_{FB}$  from cores taken from the Sea of Okhotsk, and Gulf
- of California, because the low abundances of foraminifera were utilized for other analysis. The
- analysis of bulk sediments allows for high-resolution records. Prior to the  $\delta^{15}N_{\text{bulk}}$  measurements,
- the Total Nitrogen (TN%) content of 20 sediment samples from the Sea of Okhotsk and 54 samples
- from the Gulf of California were measured at the Institute for Geology, Hamburg University,
- Germany using a flash combustion method with a Eurovector EA-3000 analyzer. The  $\delta^{15}N_{bulk}$
- measurements for both the Sea of Okhotsk and the Gulf of California were accomplished at the
- Max Planck Institute for Chemistry (Mainz), Germany using a DELTA V ADVANTAGE Isotope
- Ratio Mass Spectrometer (IRMS) equipped with a FLASH 2000 Organic Elemental Analyzer. The
- results were expressed in standard  $\delta$ -notation (equation 4). The standard deviation ( $\pm$ SD) of all
- individual analysis runs based on a certified international reference standard (USGS65) and
- internal laboratory standards (L-Phenylalanine and L-Glutamic acid) referenced to certified
- international reference standards was < 0.3%. The  $\delta^{15}N_{bulk}$  data for the Sea of Okhotsk and the
- Gulf of California are shown in Supplementary Table ST1.

$$\delta^{15}N (\%) = [(^{15}N:^{14}N_{sample})^{15}N:^{14}N_{air}) -1] \times 1,000$$
 (4)

- For the Gulf of Guayaquil core M77/2-59-01, the  $\delta^{15}N_{bulk}$  data published by Mollier-Vogel et al.
- (2019) was used. Their measurements were done on ~ 5–50 mg of homogenized and freeze-dried
- bulk sediments using a Carlo-Erba CN analyzer 2500 interfaced directly to a Micromass-Isoprime

- mass spectrometer at Bordeaux University. Results are expressed in standard  $\delta$ -notation (equation
- 4) relative to atmospheric dinitrogen gas  $(N_2)$ .
- 2.4 Nitrate offset to present conditions. The reconstructed [NO<sub>3</sub>]<sub>BW</sub> from each location is
- subtracted from the modern [NO<sub>3</sub>] present at the respective locations from similar water depths
- the cores were retrieved from. This provided the  $[NO_3^-]$  offset which is the difference ( $\Delta[NO_3^-]$
- ( $\mu$ M)) between the modern [NO<sub>3</sub><sup>-</sup>] and the past reconstructed [NO<sub>3</sub><sup>-</sup>]<sub>BW</sub>. The modern [NO<sub>3</sub><sup>-</sup>] for
- each location was taken from World Ocean Atlas 2018 (Garcia et al., 2019). The details are given
- in Supplementary Information.
- **270 3. Results**
- We reconstructed deglacial [NO<sub>3</sub>]<sub>BW</sub> using downcore sediment samples from the Sea of Okhotsk
- (MD01-2415), the Gulf of California (DSDP- 480), the Mexican Margin (MAZ-1E-04), and the
- Gulf of Guayaquil (M77/2-59-01). The reconstructed [NO<sub>3</sub><sup>-</sup>]<sub>BW</sub> was compared to  $\delta^{15}$ N<sub>bulk</sub> records
- of all cores (Fig. 4). All data records presented cover the time period starting from the Last Glacial
- Maximum, except for the core from the Sea of Okhotsk, which covers the late deglacial to the
- Holocene.
- 3.1 Sea of Okhotsk (MD01-2415). The Sea of Okhotsk core MD01-2415 covers the Younger
- Dryas, (YD, 12.8 ka BP) until the Middle to Late Holocene (MLH, 4.9 ka BP). The reconstructed
- [NO<sub>3</sub>]<sub>BW</sub> values range from 32.8 µmol/kg to 44.1 µmol/kg (Fig. 4a). A gradual increase in
- [NO<sub>3</sub>]<sub>BW</sub> is observed from the Younger Dryas to the Middle to Late Holocene. At the beginning
- of the Younger Dryas at 12.8 ka BP, [NO<sub>3</sub>]<sub>BW</sub> were relatively high and then decreased to a
- minimum value of 32.8 µmol/kg at 12.4 ka BP. Since then, [NO<sub>3</sub><sup>-</sup>]<sub>BW</sub> steadily increased until the
- 283 Middle to Late Holocene (MLH, 44.1 μmol/kg) (Fig. 4a). The [NO<sub>3</sub>]<sub>BW</sub> during the Middle to Late
- Holocene (mean =  $41.2 \mu mol/kg$ ) is significantly (t-test, p = 0.023) higher than during the Younger
- Dryas (mean = 36.7  $\mu$ mol/kg). The sedimentary  $\delta^{15}N_{bulk}$  record covers the interval from the Late
- Heinrich Stadial 1 (H1, 15.4 ka BP) to the Middle Holocene (6.1 ka BP). The  $\delta^{15}N_{bulk}$  values were
- relatively high ranging from 7.1% to 9.4% with an average of 8.7%. The  $\delta^{15}N_{bulk}$  values increased
- steadily from the Late Heinrich Stadial 1 (15.4 ka BP) to the Early Holocene (EH, 10 ka BP) with
- higher values centered between the Late Younger Dryas (11.9 ka BP) and the beginning of the
- Early Holocene. Since then, the  $\delta^{15}N_{bulk}$  values decreased until the Middle to Late Holocene.

**3.2 Gulf of California (DSDP-480)**. The analyzed sections of DSDP Site 480 covered the Last 291 Glacial Maximum (22 ka BP) until the Early Holocene (10.8 ka BP). The reconstructed [NO<sub>3</sub>]<sub>BW</sub> 292 293 ranged from 41.4 μmol/kg to 49.1 μmol/kg. The highest [NO<sub>3</sub>]<sub>BW</sub> of 49.1 μmol/kg occurred during the Last Glacial Maximum (18.2 ka BP). The data points from the Early Holocene (11.6-294 10.8 ka BP) were the only Holocene data from this core providing the lowest [NO<sub>3</sub>]<sub>BW</sub> estimate 295 of 42.1 µmol/kg during the Early Holocene (10.8 ka BP) (Fig. 4b). A distinct difference in 296 297 [NO<sub>3</sub>]<sub>BW</sub> between the glacial period (mean = 46.1 µmol/kg) and the Early Holocene (42.7 μmol/kg) was observed with [NO<sub>3</sub><sup>-</sup>]<sub>BW</sub> found to be substantially higher during the glacial period 298 (t-test, p = 0.0067) (Fig. 4b). Accordingly, the  $[NO_3^-]_{BW}$  followed a decreasing pattern from the 299 glacial period to the Early Holocene. The  $\delta^{15}N_{bulk}$  values varied between 6.4% and 13% with an 300 average of 10.2% (Fig. 4b). The  $\delta^{15}N_{bulk}$  values from the Guaymas Basin were similar to the 301 δ<sup>15</sup>N<sub>bulk</sub> values (average 9.6‰) of Pride (1997) and Altabet et al. (1999). During the last glacial 302 period, the  $\delta^{15}N_{bulk}$  values were low ranging from 8.5% to 9%. At the onset of the deglaciation, 303 the δ<sup>15</sup>N<sub>bulk</sub> values increased by more than 2% with large-scale changes reaching a maximum of 304 13% during the Younger Dryas. Afterward, we observed a gradual decline in  $\delta^{15}N_{bulk}$  values 305 306 throughout the Middle to Late Holocene (mean 10.7‰) and this pattern continued to the present. 3.3 Mexican Margin (MAZ-1E-04). This core MAZ-1E-04 covered the Last Glacial Maximum 307 308 (20.5 ka BP) until the Early Holocene (10.47 ka BP). The [NO<sub>3</sub>]<sub>BW</sub> values range from 37.7 309 μmol/kg to 43.5 μmol/kg. We observed the highest [NO<sub>3</sub>]<sub>BW</sub> during the Younger Dryas. From the 310 beginning to the end of the Last Glacial Maximum, [NO<sub>3</sub>]<sub>BW</sub> followed a decreasing trend (Fig. 4c). The [NO<sub>3</sub>]<sub>BW</sub> levels continued to steadily decrease until Heinrich Stadial 1 and consistently 311 312 stayed low throughout this period. There was a strong change in [NO<sub>3</sub><sup>-</sup>]<sub>BW</sub> from the end of Heinrich Stadial 1 to the end of Younger Dryas (Fig. 4c). We observed a peak in [NO<sub>3</sub>]<sub>BW</sub> from the 313 314 beginning of Bølling-Allerød, BA (14.29 ka BP) and it continued throughout the Younger Dryas (Fig. 4c). Afterwards,  $[NO_3^-]_{BW}$  declined during the Early Holocene. The  $\delta^{15}N_{bulk}$  values taken 315 from Alcorn et al. (2025) followed an increasing trend from the glacial towards the deglacial period 316 (Fig.4c). 317 318

3.4 Gulf of Guayaquil (M77/2-59-01). This core covered the Last Glacial Maximum (18 ka BP)
until the Middle to Late Holocene (0.18 ka BP). The reconstructed [NO<sub>3</sub><sup>-</sup>]<sub>BW</sub> values range from
40.5 μmol/kg to 46.5 μmol/kg. The highest [NO<sub>3</sub><sup>-</sup>]<sub>BW</sub> occurred during the Last Glacial Maximum
(Fig. 4d). The reconstructed [NO<sub>3</sub><sup>-</sup>]<sub>BW</sub> levels during the Last Glacial Maximum (mean = 45.6)

μmol/kg) were slightly higher than during the Middle to Late Holocene (mean = 44.9 μmol/kg) (test, p = 0.046). The  $\delta^{15}N_{bulk}$  values were relatively low ranging between 4‰ and 6‰ (Fig. 4d). During the Last Glacial Maximum, the  $\delta^{15}N_{bulk}$  values were low, varying between 4.4‰ and 4.6‰, close to the typical mean range of dissolved nitrate in the ocean (Sigman et al., 1997). Subsequently, the  $\delta^{15}N_{bulk}$  values increased from 16.7 ka BP (4.9‰), where we observed a decline in [NO<sub>3</sub> $^-$ ]<sub>BW</sub> to 8.9 ka BP (5.6‰). The highest  $\delta^{15}N_{bulk}$  values centered at ~14 ka BP (5.9‰). From 8.9 ka BP onwards, a long-term decrease in  $\delta^{15}N_{bulk}$  (< 4.4‰) was observed until the Latest Holocene, consistent with higher [NO<sub>3</sub> $^-$ ]<sub>BW</sub> levels during the Holocene (Fig. 4d). Despite higher [NO<sub>3</sub> $^-$ ]<sub>BW</sub> levels, our reconstruction doesn't show any strong variations during the Holocene.

Figure 4. Quantitative  $[NO_3^-]_{BW}$  reconstruction using the pore density of fossil specimens of *B. spissa*, *B. subadvena* from a) the Sea of Okhotsk (MD01-2415), b) the Gulf of California (DSDP-480), c) the Mexican Margin (MAZ-1E-04), and d) Gulf of Guayaquil (M77/2-59-01). The sedimentary nitrogen isotope ( $\delta^{15}N_{bulk}$ ) records from the Sea of Okhotsk, and the Gulf of California are measured in this study, and the Gulf of Guayaquil is from Mollier-Vogel et al. (2019), and the

Mexican Margin δ<sup>15</sup>N<sub>bulk</sub> data is from Alcorn et al. (2025). The error bars of [NO<sub>3</sub><sup>-</sup>]<sub>BW</sub> represent 1 SEM including a complete error propagation (using equations 3 and 4). The accumulation rate of total organic carbon (Supplementary files) calculated from published literature (Bubenshchikova et al., 2015; Leclaire & Kerry, 1982; Mollier-Vogel et al., 2019) is shown in blue dashed lines for the Sea of Okhotsk, the Gulf of California and the Gulf of Guayaquil cores respectively. The black dashed lines indicate the modern nitrate concentration of each location. Time intervals Middle to Late Holocene (MLH), Early Holocene (EH), Younger Dryas (YD), Bølling-Allerød (BA), Heinrich Stadial 1 (H1), and Last Glacial Maximum (LGM) are shown in the figure.

## 4. Discussion

- The pore density of benthic foraminifera represents a promising but still developing proxy for reconstructing past nitrate dynamics. Like most proxies based on biology, it reflects an indirect physiological response rather than a direct measure of nitrate. In addition, species-specific variability requires careful taxonomic control or its interpretation carries inherent limitations especially since not many records are available, yet, for this proxy. Thus, we used a multiproxy approach and combined it with  $\delta^{15}N_{bulk}$ , which provides a complementary perspective that strengthens reconstructions of nitrogen-cycling processes in oxygen-deficient zones.
- **4.1 Sea of Okhotsk.** Our data show that [NO<sub>3</sub>]<sub>BW</sub> levels gradually increased through time and reached modern concentrations during the Middle-Holocene (Fig. 4a). Most of the nutrients in the northwestern Pacific including the Sea of Okhotsk are supplied by the upwelling of the North Pacific Deep Water (NPDW) (Gorbarenko et al., 2014). The weakened Kuroshio current (Ujiié and Ujiié, 1999) and increased sea ice extent (Ternois et al., 2001) weakened the upwelling of NPDW during the Last Glacial Maximum (LGM). Subsequent studies (Gray et al., 2020; Rae et al., 2020) have shown that the expansion of the North Pacific Gyre also resulted in less upwelling of NPDW during the LGM.
- During the LGM, the subpolar North Pacific was better ventilated at intermediate depths (Keigwin, 1998) and export productivity was reduced (Ternois et al., 2001; Narita et al., 2002; Seki et al., 2004). This is consistent with a strengthened meridional overturning circulation, with enhanced formation of intermediate waters and advection of nutrient-depleted subtropical waters to high latitudes (Rae et al., 2020). Furthermore, the North Pacific subpolar gyre extended ~3° further

south during the LGM (Gray et al., 2020), which shifted the westerly winds southward. This may 384 have resulted in less upwelling of the NPDW during the LGM. 385 386 The prolonged ice cover with low biological productivity (Ternois et al., 2001; Narita et al., 2002; Seki et al., 2004; Rae et al., 2020) and well-oxygenated water masses (Keigwin, 1998) might have 387 prevented the formation of an oxygen deficient zone (ODZ) in the Sea of Okhotsk 388 389 (Bubenshchikova et al., 2015). This is supported by the absence of B. spissa, which are adapted to living in dysoxic conditions, in our records during the LGM. 390 Deglacial low  $[NO_3^-]_{BW}$  which correspond to higher  $\delta^{15}N_{bulk}$  values (Fig. 4a) could be due to 391 enhanced primary productivity. It is important to note, however, that  $\delta^{15}N_{bulk}$  is influenced by 392 diagenetic alteration and the incorporation of allochthonous nitrogen, which can obscure the local 393 denitrification signal. Therefore, interpretations of  $\delta^{15}N_{bulk}$  trends should be made cautiously and 394 395 ideally corroborated with complementary proxies, such as foraminiferal pore density. Increased nutrient supply from the Asian continental shelves and sea-ice retreat (Ternois et al., 2001) 396 397 strengthened primary productivity. Indeed, the accumulation rate of total organic carbon was relatively higher during the Younger Dryas (Bubenshchikova et al., 2015) in our core (Fig. 4a). 398 399 The increased oxygen demand and weakened ventilation of intermediate waters in the subarctic 400 Pacific (Lembke-Jene et al., 2018) gradually intensified the ODZ. These poorly oxygenated conditions conceivably strengthened denitrification, resulting in low deglacial [NO<sub>3</sub><sup>-</sup>]<sub>BW</sub> levels. 401 402 However, during the Middle to Late Holocene (MLH) a reorganization in atmospheric circulation 403 favored enhanced formation of oxygenated North Pacific Intermediate Water (NPIW) (Wang et al., 2020). Thus, mid-depth ventilation was closely associated with atmospheric circulation in the 404 405 Holocene and a weakened ODZ (Ohkushi et al., 2013; Bubenshchikova et al., 2015; Wang et al., 2020). These rising oxygen concentrations probably reduced denitrification (low  $\delta^{15}N_{bulk}$ ) in the 406 407 Sea of Okhotsk, resulting in higher [NO<sub>3</sub>]<sub>BW</sub> comparable to today's conditions (Fig. 4). The  $\delta^{15}N_{bulk}$  values show a maximum from 13 ka to 10 ka BP, which indicates increased water-column 408 denitrification during that time. Nevertheless, the [NO<sub>3</sub><sup>-</sup>]<sub>BW</sub> increased during this time, which 409 410 indicates a decoupling from denitrification in the oxygen minimum in the water column and the [NO<sub>3</sub>]<sub>BW</sub>. This could be related to the sea level rise during that time (Waelbroeck et al., 2008), 411 which increased the vertical distance of the sediments (i.e., bottom water) at the sampling site from 412 the center of denitrification. 413

water properties, similar to that of the open Pacific Ocean. Thus, the ODZ intensity in the Guaymas 415 416 Basin is largely dependent on the oxygen content and ventilation of inflowing NPIW from the Sea 417 of Okhotsk (Pride et al., 1999) and the demand for oxygen at depth. During the glacial period, the dissolved oxygen concentrations were higher due to better-ventilated NPIW at intermediate depths 418 419 of the Northeast Pacific (Keigwin and Jones, 1990; Ganeshram et al., 1995; Keigwin 1998; Duplessy et al., 1988; Herguera et al., 2010; Cartapanis et al., 2011). Modeling studies show that 420 421 the Laurentide and Cordilleran ice sheets increased in size (Benson et al., 2003), lowering the 422 temperature of North America (Romanova et al., 2006) during the glacial period. The cold sinking air over the ice sheet established a semi-permanent high-pressure cell (Kutzbach & Wright Jr, 423 1985; Romanova et al., 2006) causing a substantially weaker North Pacific High (Ganeshram and 424 425 Pedersen, 1998) or the southward displacement of the Inter Tropical Convergence Zone (Cheshire and Thurow, 2013). This resulted in a weak California Current along the coast and reduced 426 427 upwelling-favorable winds (Cartapanis et al., 2011) along the North American coastline and reduced primary productivity (Ganeshram and Pedersen, 1998; Hendy et al., 2004; Cartapanis et 428 429 al., 2011; Chang et al., 2015) within the ETNP and the Gulf of California during the glacial period. 430 The nitrogen isotope ratios in the Guaymas Basin can be affected by subsurface denitrification in 431 the Gulf and in the ETNP (Pride et al 1999). The increase in dissolved oxygen during the glacial period might have reduced water column denitrification (low  $\delta^{15}N_{bulk}$ ) thereby increasing the 432 433  $[NO_3]_{BW}$  (Fig. 4b). Our study finds a declining trend in reconstructed [NO<sub>3</sub>]<sub>BW</sub> during the Early Holocene, slowly 434 approaching modern concentrations. This coincides with a maximum in  $\delta^{15}N_{bulk}$  values, suggesting 435 436 elevated denitrification. This agrees with previous studies in the ETNP (Kienast et al., 2002) and 437 within the Gulf of California (Pride et al., 1999), which showed that high denitrification most likely 438 was associated with warming temperatures that occurred during this period. Furthermore, the scarcity of benthic foraminifera after the Early Holocene in our study coincides with laminations 439 of the sediment core (Keigwin & Jones, 1990) below 10.8 ka BP, where reconstructed [NO<sub>3</sub>]<sub>BW</sub> 440 begins to decrease. It is possible that redox conditions were too hostile for benthic foraminifers in 441 442 the time periods when laminated sediments formed. We acknowledge limitations in our Holocene reconstruction due to the low abundance of B. subadvena and the limited calibration dataset 443 available for this species, which may introduce a systematic offset (Govindankutty Menon et al., 444

**4.2 Gulf of California.** The Gulf of California ODZ is influenced by both intermediate and deep-

2023). Bolivina subadvena was used in this core due to the unavailability of B. spissa, and some 445 values fall outside the existing calibration range. We also cannot rule out other factors influencing 446 447 the proxy signal, such as microhabitat variability. Additional data and further proxy calibration are therefore essential to improve the robustness of Holocene bottom-water nitrate reconstructions. 448 **4.3 Mexican Margin.** Our study finds a steep rise in [NO<sub>3</sub>]<sub>BW</sub> between the Bølling-Allerød (BA) 449 450 and the Younger Dryas (YD) (Fig. 4c). The transition period from the BA to the Holocene involved rapid oxygenation with increased oxygen levels at the onset of the YD (Jaccard & Galbraith, 2012; 451 Ohkushi et al., 2013; Taylor et al., 2017). This has been linked to active ventilation by increased 452 NPIW production at high latitudes in the North Pacific (Van Geen et al., 1996; Emmer and Thunell, 453 2000; Okazaki et al., 2010; Cartapanis et al., 2011; Chang et al., 2014). In addition, there was low 454 primary productivity (Hendy et al., 2004; Pospelova et al., 2015), and a higher influx of freshwater 455 456 (Broecker et al., 1985; Clark et al., 2002) during the YD. However, considering the deep location of the Mexican Margin core below the direct influence of intermediate water masses (Fig. 3), it is 457 less likely to be reflected in the  $[NO_3^-]_{BW}$ . Bulk sediment  $\delta^{15}N$  records in the ETNP (Ganeshram 458 et al., 1995; Pride et al., 1999; Emmer and Thunell, 2000; S. S. Kienast et al., 2002; Hendy et al., 459 2004) found a decrease in  $\delta^{15}N_{\text{bulk}}$  during the YD due to reduced denitrification. Furthermore, a 460 foraminifera-bound nitrogen isotope (δ<sup>15</sup>N<sub>FB</sub>) study (Studer et al., 2021) in the eastern tropical 461 Pacific also found a decrease in  $\delta^{15}N_{FB}$  signatures during the Younger Dryas (Fig. 4c). In contrast 462 to this, a bulk sediment  $\delta^{15}N$  record of MAZ-1E-04 (Alcorn et al., 2025) depicts an increase in 463 464 water column denitrification during the Younger Dryas. Thus, reduced denitrification may not be the dominant factor that led to the elevated [NO<sub>3</sub>]<sub>BW</sub> during this time. Instead, the Mexican Margin 465 466 may be more influenced by the NO<sub>3</sub><sup>-</sup> variability from the Pacific Deep Water, PDW (see Fig. 3). Deep-sea reorganization and ventilation during the deglaciation may have influenced the 467 468 [NO<sub>3</sub><sup>-</sup>]<sub>BW</sub>. At the onset of the deglaciation, deep Southern Ocean ventilation (reduced <sup>14</sup>C 469 ventilation ages) and atmospheric carbon dioxide (CO<sub>2</sub>) synchronously increased (Robinson et al., 2009; Burke and Robinson, 2012; Rae et al., 2018). This deglacial increase in <sup>14</sup>C ventilation in 470 the Pacific Ocean suggests that most of the increase in atmospheric CO<sub>2</sub> is derived from old carbon 471 472 in the Southern and Pacific Oceans (Rafter et al., 2022). The increase in reconstructed [NO<sub>3</sub>]<sub>BW</sub> 473 during the YD may thus reflect the release of sequestered nutrient- and carbon dioxide-rich waters during the deglaciation (Robinson et al., 2009; Rafter et al., 2022). 474

The relatively high [NO<sub>3</sub>]<sub>BW</sub> during the glacial period (Fig. 4c), before its decline in Heinrich 475 476 Stadial 1, is likely indicative of reduced water-column denitrification (Ganeshram et al., 1995; 477 2000) due to reduced productivity (Ganeshram et al., 1995; Ganeshram and Pedersen, 1998) and low organic matter flux through the oxygen minimum zone (Ganeshram et al., 2000). In the ETNP, 478 including the Mexican Margin, coastal upwelling is driven by trade winds generated by subtropical 479 480 high-pressure centers. These high-pressure centers largely result from differential heating of the land and the ocean. As a result of glacial cooling on land, these high-pressure systems and the 481 associated trade winds that drive the upwelling have likely been weakened (Ganeshram and 482 Pedersen, 1998). 483 **4.4 Gulf of Guayaquil.** The core M77/2-59-01 is in a region that is sensitive to changes in 484 subsurface denitrification in the ETSP (Robinson et al., 2007, 2009; Dubois et al., 2011, 2014). 485 486 The elevated reconstructed [NO<sub>3</sub><sup>-</sup>]<sub>BW</sub> levels (Fig. 4d) during the glacial period suggest decreased water-column denitrification (Salvatteci et al., 2014; Erdem et al., 2020; Glock et al., 2022) and 487 relatively low local productivity (Ganeshram et al., 2000; Robinson et al 2007; 2009; Martinez 488 and Robinson et al., 2010; Salvatteci et al., 2016). Nutrient export to the deep Southern Ocean 489 490 waters increased due to the sluggish Atlantic Meridional Overturning Circulation (Skinner et al., 2010), and increased atmospheric iron (Fe) deposition (Somes et al., 2017) during the glacial 491 period. This reduced the transport of preformed NO<sub>3</sub><sup>-</sup> to the tropics via the Subantarctic Mode 492 493 Water (SAMW), limiting productivity. In fact, the total organic carbon (Fig. 4d) depicts low 494 productivity during this period. Furthermore, the colder sea surface temperature (SST) and the accelerated formation of SAMW and Antarctic Intermediate Water masses (Russell & Dickson, 495 496 2003; Galbraith et al., 2004) and the stronger high-latitude winds in the Southern Hemisphere 497 (Karstensen and Quadfasel, 2002) increased the ventilation rate (Meissner et al., 2005; Jaccard and 498 Galbraith, 2012; Muratli et al., 2010) during the glacial period. The resulting increased oxygen 499 concentrations (Robinson et al., 2005; Robinson et al., 2007) decreased the volume of ODZs, and nitrogen loss processes (lower δ<sup>15</sup>N<sub>bulk</sub> values, Fig. 4d) during the glacial period. In addition, 500 501 enhanced Fe deposition (Somes et al., 2017), and the glacial low sea level (Clark and Mix, 2002; 502 Wallmann et al., 2016), may have influenced the nitrate inventory in the tropical and subtropical 503 southern hemisphere. A study by Glock et al. (2018) on core M77/2-52-2 from Peru applying the pore density of B. 504 spissa also shows elevated [NO<sub>3</sub>]<sub>BW</sub> during the Last Glacial Maximum, a similar decline in [NO<sub>3</sub>] 505

BW during the Heinrich Stadial 1 and thereafter a steady decrease in [NO<sub>3</sub><sup>-</sup>]BW throughout the 506 Holocene. 507 508 The deglacial decline in [NO<sub>3</sub><sup>-</sup>] <sub>BW</sub>, especially during Heinrich Stadial 1 in this study (Fig. 4d), 509 indicates a gradual increase in surface productivity and bottom-water deoxygenation. High export production strengthened the expansion of the ETSP ODZ during the deglaciation as compared to 510 511 LGM and MLH (Salvatteci et al., 2016; Glock et al., 2018; Mollier-Vogel et al., 2019). This is consistent with the denitrification signal in the Eastern Equatorial Pacific through westward 512 513 advection from the Southeast Pacific margins (Martinez and Robinson, 2010). The shift towards generally higher reconstructed [NO<sub>3</sub>]<sub>BW</sub> from the Middle-Holocene, (Fig. 4d), 514 implies a profound change in the climatic state of the Peruvian upwelling system and the associated 515 516 ODZ during this time. From the deglaciation toward the Late Holocene, there was a general 517 increase in productivity (Mollier-Vogel et al., 2019) as shown by organic carbon accumulation rates (Fig. 4d). This increase in organic matter input and/or preservation was likely related to an 518 519 increase in upwelling-driven delivery of nutrients towards the surface. The gradual decrease in δ<sup>15</sup>N<sub>bulk</sub> values and higher [NO<sub>3</sub>]<sub>BW</sub> was likely related to a relaxation in nutrient utilization with a 520 521 nutrient supply exceeding the biological demand (Riechelson et al., 2024). Moreover, the core M77/2-59-01 was retrieved outside of the core ODZ and is under the strong influence of the oxygen 522 523 and nutrient-rich Equatorial Under Current subsurface waters (Salvatteci et al., 2019; Mollier-524 Vogel et al., 2019). These waters might have ventilated the Northern Peruvian margin and 525 deepened the oxycline at this site during the Middle-Holocene. Furthermore, enhanced zonal SST (Koutavas et al., 2006) and a northward shift of the ITCZ strengthened the Pacific Walker and 526 527 Hadley circulation during the Middle-Holocene across the tropical Pacific (Koutavas et al., 2006; 528 Mollier-Vogel et al., 2013; Salvatteci et al., 2019). These enhanced atmospheric circulations 529 brought oxygen-rich waters to intermediate depths off Peru via the equatorial subsurface 530 countercurrents (Koutavas et al., 2006; Mollier-Vogel et al., 2013; Salvatteci et al., 2019). Hence, increased ventilation of subsurface water masses reduced the strength of nitrogen loss processes 531 532 and nutrient uptake during the MLH. At present, the only quantitative reconstruction of bottom-water oxygen from these locations is the 533 core M77/2-59-01 from the Gulf of Guayaquil reported by Erdem et al. (2020). Their record 534 suggests a decline in bottom-water oxygen from the deglacial period to the Holocene. Future more 535 detailed comparisons of the nitrate reconstructions with quantitative bottom water oxygen records 536

at the same cores will further improve our understanding about variability in redox conditions and nitrogen cycling.

4.5 Comparison of past and present [NO<sub>3</sub>] at the studied locations. The [NO<sub>3</sub>] during the present and past are compared to assess the resilience of our chosen study locations towards environmental and ecological impacts of climate change. The generally positive  $\Delta[NO_3^-]$  that we found (Fig. 4b) in the Gulf of California (Guaymas Basin) and the Gulf of Guayaquil indicate that today the [NO<sub>3</sub>-] is lower than in the past. This suggests that today the nitrogen loss processes at these two core sites are stronger, most likely related to ocean warming and a decline in oxygen concentration of bottom waters. The Gulf of California core is within the heart of the oxygendeficient zone, and thus changes in ODZ oxygenation or denitrification will be more evident in this core than in any other core studied. Under nitrogen limitation, negative feedbacks (e.g., anammox) result in a decline in productivity (Naafs et al., 2019; Wallmann et al., 2022), which will stabilize the oxygen concentration. In the case of the Gulf of California, sediments are enriched in reactive iron (Fe) (Scholz et al., 2019). The decreasing NO<sub>3</sub><sup>-</sup> concentrations in the bottom water reduce the flux of NO<sub>3</sub> into the surface sediment. This leads to the release of sedimentary Fe, which enhances nitrogen fixation in the Guaymas Basin (Scholz et al., 2019). Thus, increased denitrification might not act as negative feedback in the Gulf of California because it might be countered by increased nitrogen fixation (White et al., 2013).

In the case of the Gulf of Guayaquil (Fig. 4d), whether today's elevated denitrification could enhance  $N_2$  fixation also depends on the availability of Fe (Pennington et al., 2006). The primary productivity of the Peruvian ODZ is Fe limited due to the reduction of particular Fe oxides in shelf and slope sediments (Scholz et al., 2014). Modeling studies show that primary productivity will be amplified in the Peruvian ODZ due to the release of Fe from shelf and slope sediments (Wallmann et al., 2022). This may induce deoxygenation and drive the expansion and intensification of Peruvian ODZ resulting in a positive feedback loop, like in the Gulf of California.

This situation is indicated by lower  $[NO_3^-]$  today compared to the past ~20,000 years.

The negative nitrate  $\Delta[NO_3^-]$  in the Sea of Okhotsk and the Mexican Margin (Fig. 4a & c) indicates that modern  $[NO_3^-]$  levels are higher than in the reconstructed past. This suggests that modern nitrogen loss is decreased at these two core sites compared to the last deglaciation. The higher modern  $[NO_3^-]$  in the Sea of Okhotsk is likely associated with less primary productivity and more

oxygen in the water column similar to the situation established in the MLH. The higher modern [NO<sub>3</sub>-]<sub>BW</sub> in the case of the Mexican Margin could be associated with sea level rise. The ODZ in the Mexican Continental Margin might have shifted to shallower depths today with less/or no benthic denitrification in intermediate water depths at the core site, resulting in high [NO<sub>3</sub>-]<sub>BW</sub> levels. During the glacial period, continental shelves were exposed due to sea-level lowstands (Clark and Mix, 2002; Kuhlmann et al., 2004; Wallmann et al., 2016), the main areas of primary productivity may have migrated offshore from the shallow shelf towards the continental slope relative to their Holocene positions. A similar situation occurred at the Benguela upwelling system during the LGM: TOC accumulation at the continental slope increased during the LGM in response to the seaward shift of centers of enhanced productivity (Mollenhauer et al., 2002). This offshore shift of the productivity centers and the most likely reduced remineralization rates, due to lower temperatures, indicate that the center of the ODZ at the Mexican Margin before sea level rise was possibly deeper than today. However, with the deglacial eustatic sea-level rise, the ODZ may have shifted to shallower depths. This shifted the main zone of denitrification further away from the seafloor, resulting in the increased modern [NO<sub>3</sub>-]<sub>BW</sub> in comparison to the LGM.

**5. Conclusion**. The quantitative reconstruction of [NO<sub>3</sub>]<sub>BW</sub> using the pore density of denitrifying benthic foraminifera over the last deglaciation at the four studied ODZs provides a comprehensive understanding of the past [NO<sub>3</sub>]. The Gulf of Guayaquil and Gulf of California data shows elevated [NO<sub>3</sub>]<sub>BW</sub> during the glacial period compared to deglacial and modern conditions. Considering the well-ventilated intermediate water masses in the Sea of Okhotsk, the Sea of Okhotsk may have also elevated [NO<sub>3</sub>]<sub>BW</sub> in the glacial period. For the Mexican Margin core, [NO<sub>3</sub>]<sub>BW</sub> was particularly strong during the Younger Dryas. The reconstructed [NO<sub>3</sub>]<sub>BW</sub> from the Sea of Okhotsk, the Gulf of California, and the Gulf of Guayaquil are influenced by the formation of the North Pacific Intermediate Water. However, the [NO<sub>3</sub>]<sub>BW</sub> in the deeper site, the Mexican Margin is likely influenced by the NO<sub>3</sub> variability in Pacific Deep Water. The modern Gulf of Guayaquil and the Gulf of California have low [NO<sub>3</sub>] associated with increased denitrification and a strengthening ODZ. In contrast, higher modern [NO<sub>3</sub>] was observed in the Sea of Okhotsk and the Mexican Margin, suggesting that these two study areas have higher oxygen.

#### Data availability

All data generated or analyzed during this study are available in Supplementary Information.

# **Author contributions**

- A.G.M wrote the core manuscript, did the sample preparation, electron microscopy, image and
- statistical analyses of all the fossil foraminifera. N.G. planned the study design and sampling
- strategy. G.S. hosted the research group, and provided access to SEM, and lab facilities at the
- 601 University of Hamburg. D.N. provided sampling material for cores MD01-2415 and M77/2-59-
- 602 01. C.D. provided sampling material for core MAZ-1E-04. N.L., R.S., and A.B. facilitated the
- measurement of nitrogen isotopes in the sediment samples of core MD01-2415 and DSDP-480.
- R.A. contributed to the age model development of core MAZ-1E-04, and H.F helped in the image
- processing of core DSDP-480. All authors contributed to the discussion and preparation of the
- 606 manuscript.

607

618

625

597

## **Competing interests**

- The authors declare no competing interests.
- 609 Acknowledgements
- We are grateful to the micropaleontology group at the University of Hamburg, Germany. We thank
- Alfredo Martinez-Garcia, Max Planck Institute for Chemistry, Mainz, Germany for the support
- with the measurement of nitrogen isotopes in the sediment samples. We acknowledge the help of
- Jutta Richarz, Kaya Oda for lab support, PhD student Sven Brömme (Max Planck Institute for
- 614 Chemistry, Mainz) and student assistant Hannah Krüger. We thank Yvon Balut, Agnes Baltzer,
- and the Shipboard Scientific Party of RV Marion Dufresne cruise WEPAMA 2001 for their kind
- support. We thank IODP for providing the sample for core DSDP-480. The study is a contribution
- to the Center for Earth System Research and Sustainability (CEN) of University of Hamburg.

## Financial support

- Funding was provided by the Deutsche Forschungsgemeinschaft (DFG) through both N.G.'s
- Heisenberg grant GL 999/3-1 and grant GL 999/4-1. Funding for the core MD01-2415 recovery
- was provided by the German Science Foundation (DFG) within project Ti240/11-1. The recovery
- of core M77-59 recovery was a contribution of the German Science Foundation (DFG)
- 623 Collaborative Research Project "Climate-Biogeochemistry interactions in the Tropical Ocean"
- 624 (SFB 754).

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
