# Peer review of "Quantitative reconstruction of deglacial bottom-water nitrate in marginal Pacific"

_EGUsphere, 2025_

## Author Response (AR1)

Dear Editor, Dear Reviewers,

Thank you for accepting the manuscript for the review. We were glad to receive such valuable feedbacks by both reviewers. The reviews were constructive and we think our manuscript will be improved by following the suggestions of the reviewers. Below, you can find a point-by-point response to the individual points of revision.

**Referee 1 Comments**

Thank you for giving me the opportunity to review this manuscript, entitled "Quantitative reconstruction of deglacial bottom-water nitrate in marginal Pacific seas using the pore density of denitrifying benthic foraminifera". The paper is well written and very pleasant to read, but I would suggest shortening it as it is very long and we tend to get a bit lost in the results and discussion sections. Overall, the manuscript is of significant interest in this field of research, even If I usually question the quality of the reconstruction, either O2 or nitrate, based on benthic foraminifera porosity. I have developed all my concerns below, overall I encourage publication after minor revisions.

**Reply:** Thank you very much for your detailed and helpful feedback on our manuscript. We will correct and update the manuscript accordingly, and we have provided a point-by-point response to the reviewer's minor requests and editorial suggestions below.

We have tried to provide a detailed overview of the background and environmental conditions of the different study regions, which we believe will offer a deeper understanding of the complex factors that could influence the nitrogen cycle. Taking into account the opinions of both reviewers, we will endeavor to shorten the manuscript wherever possible without compromising its essence.

**R1: L88:** So what you consider suboxic here is everything below 5 µmol/kg? Including microxic, anoxic, dysoxic? (Following Kranner et al., 2022; Hoogakker et al., 2025)

**Reply:** The giveaway in line 88, 'Denitrification (reduction of nitrate to dinitrogen gas) in the ocean occurs only in suboxic (oxygen <5 μmol/kg) conditions' (Codispoti et al., 2001; Levin, 2018), is that denitrification occurs in suboxic conditions and even harsher redox-conditions (i.e. less oxygen). That is; Nitrate loss through denitrification is increased when oxygen is depleted (Hoogakker et al., 2025). We will remove 'only' from line 88, as this could complicate and confuse the statement. Unfortunately, in many different publications so far, the exact threshold of bottom-water oxygen concentration differs.

**R1: -L93:** Do not start a sentence with an acronym\

**Reply:** Thank you for pointing this out. The following sentence will be changed in the manuscript: 'ODZs cover only 1% of the world's seafloor (Codispoti et al., 2001)' to 'Oxygen Deficient Zones cover only 1% of the world's seafloor (Codispoti et al., 2001)'.

**R1: -L113:** Maybe avoid repetition here: "The D15N records from the bulk sediment..." Plus, how about "interlinked" or something similar suggesting dependance, rather than "various"?

**Reply:** As the reviewer suggested, the sentence will be rewritten to:

The  $\delta^{15}$ N records from the bulk sediment can be subject to interlinked processes/or sources which can complicate their interpretation."

R1: -L123: "either planktic or benthic" sounds a bit strange

**Reply:** Line 122-123 "Foraminifera are unicellular eukaryotes that are abundant in marine environments (Goldstein, 1999), and can be either planktic or benthic". This will be rewritten as 'Foraminifera are unicellular eukaryotes that are abundant in marine environments' (Goldstein, 1999).

R1: -L124: less than what?

**Reply:** To clarify this, the sentence will be rewritten to: "The nitrogen isotopes of organic matter bound and protected within the calcite shell of planktic foraminifera ( $\delta^{15}N_{FB}$ ) are less subjected to diagenesis or sedimentary contamination than  $\delta^{15}N_{bulk}$  and can be used to understand major nitrogen transformations occurring in the ocean (Ren et al., 2012; Studer et al., 2021)".

R1: -L138: "Some species, for example *Bolivina spissa*, which are..."

**Reply:** Okay. We will correct the comma placement accordingly.

**R1:** -L149-152: Be cautious here, Bolivinidae contains a lot of bolivina species, on how many species has this hypothesis been tested?

**Reply:** Lines 149-152 as follows "Every *Bolivina* species tested so far (including *Bolivina seminuda*), can denitrify (Piña-Ochoa et al., 2010a; Bernhard et al., 2012b), suggesting that denitrification is a common survival strategy of Bolivinidae under oxygen-depleted conditions (Glock et al., 2019)". In total 8/8 *Bolivina* species that have been tested can denitrify and 12/12 *Bolivina* species that have been analyzed for intracellular nitrate storage, store significant amounts of nitrate (Glock, 2023). This makes species from this genus particularly suitable candidates for past nitrate reconstruction through the analysis of pore characteristics as an empirical proxy.

R1: -L152-154: Reference for this?

**Reply:** Govindankutty Menon et al., 2023 will be added as a reference.

**R1:** -L156: Bolivina in italics

**Reply:** Thank you for pointing out that. Line 156 "Therefore, the pore density of several Bolivina species....." will be changed in the manuscript as "Therefore, the pore density of several *Bolivina* species.....".

**R1:** -Figure 1: not sure I fully understand panel B: if the red cross is for B. subadvena, does it mean all the other symbol are for *B. spissa*? What location is *B. subadvena* then? Not sure about mixing species and location here.

**Reply:** The red cross depicts *Bolivina subadvena*, while the other symbols depict *Bolivina spissa*, which was collected from Peru, Costa Rica, and Sagami Bay. *B. subadvena* was collected in Costa Rica and Sagami Bay. A detailed description of all the studied locations and species along with the calibration figure can be found in Govindankutty Menon et al. (2023). To clarify this, we will add to the figure caption: "If no species name is indicated in the legend, the analysed species was *B. spissa*."

**R1:** -Figure 2: not very colorblind friendly, please use a gradient with only one color or two colorblind friendly colors;

**Reply:** Okay. We will modify the color scale of Figure 2 using that of Figure 3 in the manuscript.

R1: -Also, why is the chosen depth 700m? why not an average of your core depth for example?

**Reply:** The deepest core used in the study was taken at a water depth of 1,463 m, while the shallowest core was taken at a depth of 747 m. Given that the shallowest recorded sediment core depths are 747 m, we believe that using the 700 m surface oxygen concentrations for Figure 2 is appropriate. This is because 700 m could represent surface-to-intermediate water oxygen concentrations, and this is usually around the lower boundary of ODZs. This boundary encompasses the well-ventilated upper ocean while excluding the deeper, less oxygenated water masses typically found below ~700 m in modern oceanographic profiles. The 700 m threshold is also above the core of regional oxygen minimum zones and corresponds to the base of most seasonal and atmospheric influences on dissolved oxygen.

**R1:** -Figure 3: I recommend using this color scale for Fig. 2.

Reply: Done.

**R1:** -Also, we are still in the introduction on page 8, and there are already 3 figures. It might be nice to try to shorten the manuscript a bit.

**Reply:** Okay. We will try our best to shorten the manuscript, wherever it is possible.

R1: -L201 to 214: please check between RV and R/V

**Reply:** Thank you for pointing out that. RV will be consistently used in lines L201 to 214 in the manuscript.

**R1:** L215-220: to bit hard to read and to follow. I would just write "A total of 1541 specimens were investigated (1268 B. spissa and 273 B. subadvena) across X samples from location A, B, C..." without detailing too much. You can also put these information in a supplementary table for

example., but I'm not sure all the information given in this paragraph are very important in the end.

**Reply:** We will move section 2.2 into the supplementary information and adjust the sentence according to the suggestions by the reviewer.

**R1:** -L229: Do not detail everything you did not do.

**Reply:** Okay. We will remove line 229 "The specimens were not sputter-coated to allow for future geochemical analyses" from this part.

**R1:** -L230: I am quite surprised that you worked with the oldest chambers, as porosity investigated is usually performed on the latest chambers (penultimate or antepenultimate) to avoid too much pore calcification, infilling, pore plate, which are quite common in Bolivina species, can you justify this choice?

**Reply:** This is an extremely important aspect to address for. The first (oldest) ten chambers are used for the porosity measurements in order to reduce the ontogenetic effect on the pore density and porosity of chambers. This can be seen on **Fig. 1a**. The pore density is increasing with every subsequent chamber. If the more recent 1-2 chambers would be used, then we would need to use the specimens within the same ontogenetic stage. That is, the size of the specimen and the number of chambers should be the same in all the chosen specimens. It is nearly impossible to find many specimens having exactly the same number of chambers. Moreover, broken fragments cannot be traced back to a specific chamber. By sticking to the oldest chambers of the foraminifer the ontogenetic effects are minimized by size normalization. Moreover, the larger area provides a statistically robust, and larger dataset for each analyzed specimen. The stability of the test walls is less restricting since the older parts of the test are usually less porous (Glock et al., 2022). However, we do need to mention that this method requires greater effort to acquire the data. We have already added the reason behind the selection of oldest chambers in Govindankutty Menon et al., 2023. Nevertheless, we could add into the supplementary information of the current manuscript.

**R1:** -L233: Can you develop what are pore density, porosity, and so on?

**Reply:** The pore density is the number of pores per unit area, and the porosity is the percentage of the area of the tests occupied by the pores. The mean pore size is the average pore sizes of one individual. We believe that adding these definitions to the manuscript will increase its length. Therefore, we would like to include these definitions in the supplementary information.

**R1:** -L234: previously trained deep learning algorithm: reference for that?

**Reply:** The reference for the previously trained deep learning algorithm is Govindankutty Menon et al. (2023). We will add this to the manuscript at this point.

**R1:** -L235: this sentence is not necessary

**Reply:** Okay. We will remove line 235 "The deep learning algorithm that was used for this study is included in the Amira software package".

**R1-**L288: Overall in paragraph 2.4, why are you sometimes using Intcal20, sometimes Marine20?

**Reply:** IntCal20 was selected for core MAZ-1E-04 to maintain consistency with Alcorn et al. (2025, preprint), who also studied this core. As their work involved detailed benthic ventilation age reconstructions, a more precise manual reservoir age correction was applied. In Marine20, a continuous reservoir age correction is done by using the changes in globally averaged reservoir ages, which is not sufficient for local ventilation age reconstructions, since the reservoir ages vary in time and space.

**R1**-Table 1: MD01-2415: It's a zero, not an "O"

Reply: Done.

**R1-**L344: We are already page 14, and only at the results section, please try to be more concise and shorten the manuscript, even if I have to admit it is very pleasant to read.

**Reply:** We will do our best to shorten the manuscript. To this end, we will move sections 2.2 and 2.4 to the supplementary information section and remove Fig. 5.

**R1-**Figure 4: I suggest to remove the big arrow on the left. By looking at this figure, I would say that the link between NO3- and D15N is not clear, or that the link between porosity and Nitrate is not clear. I tried to develop method for reconstructing O2 based on porosity and my experience led me to think that this is very tricky indeed as we try to retrieve information from the response of a single species, or here 2, that usually live in a restricted range of the parameters we are trying to investigate. Have you tried to compare the signal you extract from the porosity to existing O2 reconstruction for these core site?

In my experience, porosity reconstruction or using it as a transfer function also show a large margin of error, so I'm always very dubitative when I see new transfer function based on foraminiferal porosity.

**Reply:** We will remove the big arrow on the left from the Figure 4.

Except for the core M77/2-59-01 from the Gulf of Guayaquil, as presented by Erdem et al. (2020), we are unaware of any other quantitative reconstructions of bottom-water oxygen. Additionally, there are significant data gaps in the Erdem et al. (2020) dataset. While they suggest a decrease in bottom-water oxygen levels from the deglacial period to the Holocene, these gaps complicate direct comparisons with our reconstructed bottom-water nitrate levels. We are currently working on a manuscript that explores the redox conditions of the four cores studied, and we have plans for a follow-up study in which we will utilize the Mn/Ca ratio of foraminifera to further refine our understanding of these redox conditions.

**R1-**I think this is important as you suggest later on that O2 can also play a role, and I think it actually is. Finally, maybe put the letters a, b, c, and d on the lower part of each graph?

**Reply:** Done. The letters a, b, c, and d will be moved to the lower part of each graph.

**R1-**Legend of Fig. 4: maybe put the dashed line in black? As red is the color you used for D15N?

**Reply:** Done.

R1-L432 to 436: maybe you don't need to explain all of that?

**Reply:** This will be moved to the figure caption.

**R1-**L438: Yes, but at the same time, all your records don't have modern samples, so don't shoot you in the foot like this, unless you can argue that the Holocene shows not variability.

**Reply:** We will take out that part from the manuscript.

**R1-**Figure 5: Here would be a good place to add a comparison with O2, to try to investigate which part of your signal is actually dependent on oxygenation. Now, several reconstructions exist for these areas; If not I think this is redundant with Fig. 4 as you have no new information here, so I would suggest removing it to try to shorten the manuscript

**Reply:** Please refer to our response to the above comment regarding the comparison with oxygen. We will remove Fig. 5 from the manuscript.

**R1-**L450: what is a SEM?

**Reply:** SEM stands for 'Standard Error of the Mean'. This has already been explained at line 251 of the manuscript.

**R1**L453 to L456: is it necessary to develop that?

**Reply:** Lines453-456 will be removed from the manuscript.

**R1-**L493: again here, try to add a comparison with O2 reconstructions?

**Reply:** As of the latest available data, no quantitative bottom-water oxygen reconstructions exist for the MD01-2415 core from the Sea of Okhotsk.

**R1-**L522-524: Not sure I get it there: If the argument is using B. species for nitrate reconstruction because they rely on denitrification and don't need O2, why would they be dependent on O2 and redox conditions here? If they still do depend on O2, then this relationship should be discussed here if O2 can account for part of the porosity investigated here.

**Reply:** Indeed, *Bolivina spissa* depend on nitrate rather than oxygen for respiration (i.e., denitrification). This metabolic flexibility provides a significant survival advantage in environments with low oxygen levels, such as oxygen minimum zones (OMZs) or suboxic

sediments. This species can switch to oxygen respiration, if it has to, but it has a preference for denitrification (Glock et al., 2019) and will normally try to avoid exposure to oxygen, since trace amounts of oxygen already can inhibit denitrification. However, if the redox conditions become too harsh, even nitrate can be completely depleted from the pore water or the surrounding environment. In such strongly anoxic (euxinic) conditions, nitrate is rapidly reduced (i.e., from nitrate to nitrite and then to nitrogen gas) and becomes unavailable. Sulfate reduction and methanogenesis then become the dominant terminal electron-accepting processes (Lam & Kuypers, 2011). In this type of environmental setting, *Bolivina spissa* and other nitrate-respiring foraminifera cannot survive or function metabolically. In addition, overexposure of toxic sulfide is most likely also harmful for the foraminifera.

**R1-** Overall in the rest of the manuscript, I feel like we have these porosity reconstructions, that are supposedly related to nitrate concentration here, but I won't question that further, and then lots of supposition around these reconstructions during the whole discussion. I feel like you are not bringing argument to discuss or defend a hypothesis, but rather throwing a lots of questions around that.....

**Reply:** The strong correlation between the pore density in *B. spissa* and bottom water nitrate concentrations is not really a new finding. It goes back to the original study by Glock et al. (2011) and has been applied and compared with another quantitative nitrate proxy in Glock et al. (2018). Later, the calibration has been extended to other sampling regions and another *Bolivina* species by Govindankutty Menon et al. (2023). The N-cycle is highly complex and bottom water nitrate concentrations can be influenced by several processes, that are all discussed in the paper (e.g. transport of nutrients to the bottom water via the biological pump; denitrification in the water column within the OMZ above the sampling location; local redox conditions and denitrification in the bottom water; ...). These processes have not so well been explored into the past, yet, since there aren't that many proxies to quantitatively reconstruct bottom water nitrate concentrations, yet. Since the other proxy that is typically used to explore nitrogen cycling (d15N) is measured on organic matter that is formed in shallower water depths, it is usually not directly coupled to the bottom water N-cycling, which further complicates the situation. Rather than simply defending a single hypothesis, our study provides the reader with the opportunity to appreciate the complex dynamics of the nitrogen cycle. For the first time, we quantitatively reconstruct nitrate concentrations at four distinct locations, offering valuable insights into regional variations in bottom water nitrogen cycling. Given the increasing global reliance on nitrogenous fertilizers, we believe that continued research in this area is essential for enhancing climate predictions and understanding the broader environmental impact of nitrogen on ecosystems. The ability to quantitatively assess nitrate at multiple locations offers a new dimension to understanding the nitrogen cycle, which has been largely constrained by regionspecific data. These findings are not only pivotal for our scientific understanding but also provide a foundational basis for future research that can address critical questions in both the natural and anthropogenic nitrogen cycles.

**R1-**L667-668: again, at least a few O2 reconstructions exists for these periods of time and location, so you can use them instead of wondering if the O2 level was different.

**Reply:** Please refer to our above response "Except for the core M77/2-59-01 from the Gulf of Guayaquil, as presented by Erdem et al. (2020), we are unaware of any other quantitative reconstructions of bottom-water oxygen. Additionally, there are significant data gaps in the Erdem et al. (2020) dataset. While they suggest a decrease in bottom-water oxygen levels from the deglacial period to the Holocene, these gaps complicate direct comparisons with our reconstructed bottom-water nitrate levels. We are currently working on a manuscript that explores the redox conditions of the four cores studied, and we have plans for a follow-up study in which we will utilize the Mn/Ca ratio of foraminifera to further refine our understanding of these redox conditions".

**Reference**

Hoogakker, B. A., Davis, C., Wang, Y., Kusch, S., Nilsson-Kerr, K., Hardisty, D. S., ... & Zhou, Y. (2025). Reviews and syntheses: Review of proxies for low-oxygen paleoceanographic reconstructions. *Biogeosciences*, 22(4), 863-957.

**Referee 2 Comments**

Review of "Quantitative reconstruction of deglacial bottom-water nitrate in marginal Pacific seas using the pore density of denitrifying benthic foraminifera," submitted to Climate of the Past by Anjaly Govindankutty Menon et al.

Govidankutty Menon et al. present reconstructions of bottom water nitrate concentration based on the porosity of infaunal benthic foraminifera and bulk sediment d15N from four locations in the Pacific Ocean with intermittent data coverage over the last 30,000 years. The authors use these data to infer the complex climate and oceanographic drivers of regional oxygen deficiency over this timescale.

Overall, this reviewer found a strong core of this paper in the porosity data, a new and novel proxy for bottom water nitrate concentration. Additionally, the comparison to d15Nbulk is sound. However, I recommend major revisions at this stage in recognition of what I find to be two major weaknesses. First, the paper is groaning under the weight of the content provided, especially throughout the introduction and the discussion. I appreciate the authors providing a thorough review here, but the weight of the material reads more like a dissertation than a manuscript, and the text would be better served through a focus on the specific proxies and results of the study. As testament to this, having only five figures across seven hundred lines of text is not ideal. Much of my comments below are to help improve the manuscript's readability by shortening.

Second, the manuscript is missing a critical interpretation of the proxies employed. Specifically, the data generated (and their gaps) raise uncertainties regarding the reliability of the porosity proxy for NO3\_bw. The authors are well-positioned to address these uncertainties, but do not in the current version. Relatedly, the authors present d15Nbulk as a reliable indicator of denitrification, when this reliability has been questioned. Stepping back, I think the manuscript would be greatly improved if the authors reframe their discussion from the current detailed mechanistic interpretation of the results – which does not appear warranted at this stage, and in any case would not be necessary for publication – for a critical evaluation of their proxy systems.

**Reply:** Thank you for taking the time to review our manuscript. We will try to shorten the manuscript wherever possible. For example, sections 2.2 and 2.4 will be moved to the supplementary information, and Fig. 5 will be removed from the manuscript. We will also cut the part on foraminifera-based D15N from the introduction and add a paragraph on disagreements in existing literature regarding D15N and FB-15N. Furthermore, a small section on the limitations of pore density of benthic foraminifera will be added in the discussion part of the manuscript.

**R2:** I hope the below comments are helpful and would be happy to take a deeper dive into the interpretation of a revised manuscript.

Major comments.

**R2-** Introduction, L102-180. I commend the authors for trying to cover a lot of ground here, from d15N\_bulk to FBd15N to benthic foraminifera pore density. However, this is an extremely challenging section for the reader to follow as currently written. Some organization could help immensely. For one potential solution: After L101, first please introduce the main output of this study (for instance, the text on L169-171 and L174-180). Then have separate subsections reviewing the different proxies used (Section 1.1 on d15Nbulk, Section 1.2 on pore density). Additionally, I would suggest largely removing the section on FB-d15N given that the authors do not employ this. Instead, they should mention the known disagreements between d15Nbulk and FB-d15N in their section on d15Nbulk, with relevant references (e.g., Studer et al., 2021). The caveat on the interpretation of d15Nbulk as a proxy of ODZ intensity given potential diagenetic and allochthonous N incorporation should be made explicit and raised throughout the discussion.

**Reply:** Done. We will restructure the introduction after line 101 as follows "In this study, we use the pore density (number of pores per unit area) of *B. spissa* and *B. subadvena* as a  $NO_3^-$  proxy (Govindankutty Menon et al., 2023) to reconstruct  $[NO_3^-]_{BW}$  in intermediate water depths of the Sea of Okhotsk, the Gulf of California, the Gulf of Guayaquil, and in the Pacific Deep Water (PDW) depths of the Mexican Margin (Fig. 2 and 3). The  $[NO_3^-]_{BW}$  calibration using the pore density of *B. spissa* and *B. subadvena* (see Fig. 1 (b)) developed in Govindankutty Menon et al. (2023) is applied in the current study. Combining a proxy for bottom-water nitrate  $[NO_3^-]_{BW}$  (pore density of denitrifying foraminifera) and a proxy for N-cycle processes in the water column  $(\delta^{15}N_{bulk})$  might facilitate a more comprehensive understanding of past N-cycling in different zones of the water column. Here, we try to understand 1) whether there are differences in

reconstructed  $[NO_3^-]_{BW}$  between today, deglacial, and glacial periods in the four studied sites, and 2) whether the reconstructed  $[NO_3^-]_{BW}$  records are in agreement with insights drawn from  $\delta^{15}N_{bulk}$  data".

Then, we will continue with Section 1.1 on application of  $\delta^{15}N_{bulk}$  and its potential limitations, along with the known disagreements between  $\delta^{15}N_{bulk}$  and FB- $\delta^{15}N$  and Section 1.2 on the pore density of benthic foraminifera as a bottom-water nitrate proxy. Also, we will explicitly discuss the potential limitations of  $\delta^{15}N_{bulk}$  as a proxy of ODZ intensity in the discussion section.

**R2-**Bottom water vs. pore water [NO3-]. B. spissa is a shallow infaunal species (Glock et al., 2011), yet the authors are correlating pore density to bottom water, and not pore water, nitrate concentration. Does this difference matter, and has it been tested previously? In these low oxygen areas, it would be expected that nitrate would be rapidly consumed in the pore water by denitrification. Perhaps the authors or others have discussed this in previous papers, but it would be beneficial to have a summary here.

**Reply:** The original study by Glock et al. (2011) shows the strong correlation between the pore density in Bolivina spissa and bottom water nitrate concentrations. This has been applied and compared with another quantitative nitrate proxy presented in Glock et al. (2018). Both independently calibrated proxies showed nearly the same quantitative results. Glock et al. (2011) compared bottom-water nitrate and mean pore water nitrate within the first centimeter of the sediment. They concluded that in a hypothetical zone of nitrate uptake, the pore water nitrate uptake found to be lower than the bottom-water nitrate. While pore water nitrate can reflect bottom-water nitrate to an extent, it is subject to variability due to microbial diagenesis, organic matter input etc. In addition, infaunal foraminifera can migrate in the pore water. So, the pore density is reflecting an average of the nitrate exposure, they experience over their lifetime. Finally, Bolivina spissa is not living deep infaunal. For example, off Peru most of them are found in the top 3 mm of the sediment (Malon, 2012). Then, at the end, we have to keep in mind, that this proxy (like most palaeoproxies) is an empirically calibrated proxy, which showed the best correlation with the bottom water nitrate concentration and not bottom water oxygen, temperature, water depth, salinity or pore water nitrate (Glock et al., 2011, Govindankutty Menon et al., 2023). Of course, we cannot exclude other factors, like for example a change in microhabitat, but this also concerns many other proxies. We will add two sentences to the manuscript to summarize this to the readers.

**R2-**Age models (Section 2.4). There seems to be some unnecessary repetition here. On L282-284, the authors outline the approach of subtracting MRAs from raw 14C ages and conversion to calendar age using Intcal20. Isn't this the same approach that is applied to all cores? I'd suggest two changes to improve this section: First, please first state the commonality of the approach for all locations at the beginning of Section 2.4. Second, please save the text here by adding the 14C age source,  $\Delta R$ , and source for each core site to the existing Table 1.

**Reply:** Done. We will add the  $^{14}$ C age,  $\Delta R$  and source for each core site to the existing Table 1.

Marine20 was used to develop the age model for cores from the Sea of Okhotsk, the Gulf of California, and the Gulf of Guayaquil, based on the published 14C ages of planktic foraminifera. However, in Marine20, a continuous reservoir age correction is performed using changes in the globally averaged reservoir age. This is insufficient for local ventilation age reconstructions since reservoir ages vary over time and space. IntCal20 was selected for the Mexican Margin core to maintain consistency with Alcorn et al. (2025, preprint), who also studied this core. Since their work involved detailed benthic ventilation age reconstructions, a more precise manual reservoir age correction was applied.

As the reviewer suggested, we will shorten the revised manuscript and move Section 2.4 to the supplementary information.

**R2-**There's a fundamental challenge with the results of this study: Through no fault of the authors, the reconstructions of NO3\_bw only extend to the Late Holocene at one of their four locations. This discontinuity would lead to interpretive discomfort with any proxy, but particulary a new proxy with few paleo applications. For example, for the DSDP 480 record (Gulf of California), the authors state that "none of the values are close to the modern values" (L436). The authors go on to interpret a decline in NO3\_bw in the Gulf of California over the Holocene, but they do not have data to prove this, and do not raise the equally or more likely possibility that there is a systematic proxy offset here.

**Reply:** We understand the reviewer's concern that we don't have modern samples. Lines 436 to 437 are explained on this basis: in comparison to other cores, a large nitrate offset to modern bottom water nitrate exists for the core from the Gulf of California. Throughout the Early Holocene, the  $\delta^{15}N_{\text{bulk}}$  is increasing, which is likely related to an increase in denitrification. In addition, in core sections, where we didn't find enough B. subadvena specimens, the sediment was typically laminated, which includes most parts of the Holocene. Thus, both proxies and some other proxy records from this region within the literature that is cited in this section (Pride et al., 1999, Kienast et al., 2002), indicate that the conditions were strongly reducing during this period. The conditions today are still very harsh and studies in the water column of the Guyamas Basin show that there is a huge nitrate deficit in the water, due to local denitrification (Scholz et al., 2019) We therefore interpreted the strongly elevated nitrate concentrations during the LGM, with increased oxygen concentrations and strongly reduced denitrification activity. The difference between the modern bottom water nitrate concentrations and the reconstructed glacial concentrations is similar as the modern nitrate deficit in the water column that has been found by Scholz et al. (2019). Therefore, it is not unlikely that our pore density signal indeed represents strongly reduced water column denitrification in this region during the glacial and parts of the deglaciation. We agree that more data is needed on the Holocene to gain a concrete understanding of the trend in bottom-water nitrate levels and of course, we cannot exclude other factors, influencing the proxy results (see microhabitat variation as mentioned above). Furthermore, Bolivina subadvena was used for this core due to the unavailability of Bolivina spissa. Therefore, there may be a systematic proxy offset here, since we do not have so many calibration points for B. subadvena (Govindankutty Menon et al., 2023) and some datapoints are out of range of the calibration. Therefore, more data and proxy calibrations are necessary for a concrete analysis of the trend of bottom-water nitrate in the Holocene. Thus, Lines 437-438 will be removed from the manuscript, and a line will be included in the discussion section to address the limitations of the proxy imposed by our lack of Holocene data. In addition, we will add a line about the potential uncertainties due to the use of *B. subadvena* and potential microhabitat changes during the Glacial.

Line by line comments through the results.

**R2**-L84. Please specify "very low dissolved oxygen".

**Reply:** Line 84 will be reformulated as "Oxygen-deficient zones (ODZs) are regions of very low dissolved oxygen (O2)".

**R2-**L91-93. This sentence is difficult to follow because it is covering a huge amount of territory. Please simplify/revise.

**Reply:** Line 91-93 can be reformulated as follows "Due to the complex interactions and feedbacks within the biogeochemical nitrogen cycle, the amount of benthic denitrification also influences other important processes, such as global nitrogen fixation and net primary production (Somes et al., 2017; Li et al., 2024)".

**R2**-L97-98. This statement is true regarding climate models, however it does not mention the growing consensus from the paleo literature that the ODZs contract during both transient and equilibrium climate warmings on a timescale of millennia and longer (Auderset et al., 2022 and Moretti et al., 2024). Consider rephrasing and including these references here.

**Reply:** Done. We will modify line 95 onwards like this "Observations and climate model simulations have predicted that ODZs will continue to expand until at least the year 2100 (Stramma et al., 2008, 2010; Schmidtko et al., 2017; Oschlies, 2021). However, the long-term evolution of ODZs remains uncertain (Yamamoto et al., 2015; Takano et al., 2018; Fu et al., 2018; Frölicher et al., 2020). There is a growing consensus that ODZs contract during transient and equilibrium climate warmings over timescales of millennia and beyond (Auderset et al., 2022; Moretti et al., 2024)".

**R2**-L107. Add "the" – "when the oxygen supply"

**Reply:** Done.

**R2**-L107-110. This statement is a bit all over the map. Please consider rephrasing in terms of the oxygen availability in the subsurface ODZ regions. This availability is a combination of oxygen supply (driven both by air-sea equilibration and interior ocean ventilation) and oxygen demand (both at the core site and also along the path of the ventilating watermass).

**Reply:** Done. Lines 107-110 will rephrased as "When the oxygen in the ocean is depleted, either due to global warming or increased remineralization, denitrification rates in the water column are also increasing and so is  $\delta^{15}N$  (Wang et al., 2019)".

**R2**-L137. What is "a large fraction"? Please specify approximate range from these studies (in percentage).

**Reply:** By "a large fraction" we meant that a major part of benthic denitrification in the ODZs is by foraminifera. For the ease of understanding, we will reformulate **Line137 as** "Foraminifera account for a major part of benthic denitrification in the ODZs (up to 100% in some environments) (Piña-Ochoa et al., 2010a; 2010b; Glock et al., 2013; Dale et al., 2016, Chocquel et al., 2021, Rakshit et al., 2025).

**R2**-L177-179. This can be simplified – the authors are "try[ing] to understand 1) whether there are differences in reconstructed NO3\_bw between today, deglacial, and glacial periods in the four studied sites, and 2) whether the reconstructed NO3\_bw records are in agreement with insights drawn from d15N\_bulk data".

**Reply:** Done.

R2-L219. ", and 11 sample depths"

**Reply:** Done.

**References**

Choquel, C., Geslin, E., Metzger, E., Filipsson, H. L., Risgaard-Petersen, N., Launeau, P., ... & Mouret, A. (2021). Denitrification by benthic foraminifera and their contribution to N-loss from a fjord environment. *Biogeosciences*, *18*(1), 327-341.

Glock, N. (2023). Benthic foraminifera and gromiids from oxygen-depleted environments—survival strategies, biogeochemistry and trophic interactions. *Biogeosciences*, 20(16), 3423-3447.

Mallon, J. (2012). *Benthic foraminifera of the Peruvian and Ecuadorian continental margin* (Doctoral dissertation).

Rakshit, S., Glock, N., Dale, A. W., Armstrong, M. M., Scholz, F., Mutzberg, A., & Algar, C. K. (2025). Foraminiferal denitrification and deep bioirrigation influence benthic biogeochemical cycling in a seasonally hypoxic fjord. *Geochimica et Cosmochimica Acta*, 388, 268-282.